# Temporal–Spatial Evolution of Kinetic and Thermal Energy Dissipation Rates in a Three-Dimensional Turbulent Rayleigh–Taylor Mixing Zone

**DOI:** 10.3390/e22060652

**Published:** 2020-06-12

**Authors:** Wenjing Guo, Xiurong Guo, Yikun Wei, Yan Zhang

**Affiliations:** 1Basic Courses Department, Shandong University of Science and Technology, Taian 271019, China; skd993808@sdust.edu.cn (W.G.); skd991314@sdust.edu.cn (X.G.); 2State-Province Joint Engineering Lab of Fluid Transmission System Technology, Faculty of Mechanical Engineering and Automation, Zhejiang Sci-Tech University, Hangzhou 310018, China; 3Department of Aeronautics, Imperial College London, London SW72AZ, UK

**Keywords:** Rayleigh–Taylor, energy dissipation, heat transport, lattice Boltzmann method

## Abstract

In this work, the temporal–spatial evolution of kinetic and thermal energy dissipation rates in three-dimensional (3D) turbulent Rayleigh–Taylor (RT) mixing are investigated numerically by the lattice Boltzmann method. The temperature fields, kinetic and thermal energy dissipation rates with temporal–spatial evolution, the probability density functions, the fractal dimension of mixing interface, spatial scaling law of structure function for the kinetic and the thermal energy dissipation rates in 3D space are analysed in detail to provide an improved physical understanding of the temporal–spatial dissipation-rate characteristic in the 3D turbulent Rayleigh–Taylor mixing zone. Our numerical results indicate that the kinetic and thermal energy dissipation rates are concentrated in areas with large gradients of velocity and temperature with temporal evolution, respectively, which is consistent with the theoretical assumption. However, small scale thermal plumes initially at the section of half vertical height increasingly develop large scale plumes with time evolution. The probability density function tail of thermal energy dissipation gradually rises and approaches the stretched exponent function with temporal evolution. The slope of fractal dimension increases at an early time, however, the fractal dimension for the fluid interfaces is 2.4 at times *t*/*τ* ≥ 2, which demonstrates the self-similarity of the turbulent RT mixing zone in 3D space. It is further demonstrated that the second, fourth and sixth-order structure functions for velocity and temperature structure functions have a linear scaling within the inertial range.

## 1. Introduction

The Rayleigh–Taylor (RT) instability phenomenon is of great importance to various fields of science, technology, and astrophysics [1,2,3,4]. The full turbulent nonlinear phenomena of RT instability occur in various natural systems with unstable interfaces [5]. The RT instability is mainly derived by two layers (heavier–lighter layers, or colder–hotter layers) of a single-phase fluid due to the appearance of relative acceleration [6,7]. To deeply understand the transport characteristics of both the kinetic energy dissipation rate (ε_u_) and thermal energy dissipation rate (ε_θ_) inside the mixing zone is among the essential problems in turbulent RT instability. Chertkov (2003) [6] proposed a statistical properties theoretical model of $\varepsilon_{u}$ and $\varepsilon_{\theta }$ for two-dimensional (2D) and three-dimensional (3D) RT turbulence. 

In the 3D Chertkov model, the Kolmogorov (K41) scenario is proposed for velocity and temperature spectra [6]. In the 2D Chertkov model [6], the Bolgiano–Obukhov-like (BO59) scaling appears due to the fully active temperature [8,9]. The expressions of kinetic and thermal energy dissipation rates are given as [6].
(1)$$\varepsilon_{u}(x,t)\equiv 0.5\nu {\sum_{i,j}\left[\frac{\partial u_{j}(x,t)}{\partial x_{i}}+\frac{\partial u_{i}(x,t)}{\partial x_{j}}\right]}^{2}$$
and
(2)$$\varepsilon_{\theta }(x,t)\equiv \kappa {\sum_{i}\left[\frac{\partial \theta (x,t)}{\partial x_{i}}\right]}^{2}$$
where ν denotes the kinematic viscosity of the fluid, u is the velocity of the fluid, θ is the temperature, and κ is the thermal diffusivity of the fluid. The kinetic and thermal energy dissipation rates can be among the most important topics in turbulent RT instability. For instance, Dalziel et al. (1999) [9] and Biferale et al. (2010) [10] reported the validity of the B59 scenario by direct numerical simulation (DNS) in 2D and using a thermal lattice Boltzmann method (LBM) in 3D, respectively. Zhou et al. (2013,2016) [11,12,13] and Qiu et al. [14] have received strong support for the velocity and temperature B59 scaling of RT turbulence from both in space and in time in the 2D case. Zhou et al. (2016) [12] showed statistical properties of the kinetic and thermal energy dissipation rates in 2D RT turbulence by means of direct numerical simulations (DNS). They argued that intense dissipation events occur near the interfaces of hot and cold fluids between the kinetic and thermal energy dissipation rates, leading to a strong positive correlation in the turbulent range [12]. In the 3D RT turbulence, this theory proposed by Chertkov (2003) [6] predicts a Kolmogorov-like scenario, with a quasi-stationary energy cascade in the mixing layer. Its prediction is in view of the Kolmogorov–Obukhov picture of turbulence where fluctuations of density are passively transported in the cascade [15,16,17]. 

Once the statistical properties of the kinetic and thermal energy dissipation rates in the 2D RT turbulence have been described, in the 2D case, the kinetic and thermal dissipation rates with only one plane are represented in the turbulent RT mixing evolution. However, new open questions will emerge in 3D. How do we understand the kinetic and thermal energy dissipation rates caused by the velocity gradient of space and temperature gradient of space inside the RT mixing zone? If the above main factors are considered in 3D to affect the kinetic and thermal energy dissipation rates, the physical insight of understanding the kinetic and thermal energy dissipation rates in 3D are significant.

Based on the above discussions, we mainly focus on the statistical properties of the temporal–spatial evolution of kinetic and thermal energy dissipation rates in 3D turbulent Rayleigh–Taylor mixing with the aim of deepening and extending previous investigation in 2D [12]. As seen in the above discussions, there is a temperature fluctuation as an active scalar in 3D turbulent RT mixing and thus a Kolmogorov-like scenario [18,19]; the non-BO59 scaling was theoretically predicted [20,21] and numerically demonstrated [22,23,24]. We mainly extend the previous theoretical prediction [6] and numerical investigations [12] in a 2D RT system. Our numerical results mainly reveal that several insight differences in the statistical properties of kinetic and thermal energy dissipation rates can be obtained in 3D turbulent Rayleigh–Taylor mixing compared to what is observed previously in 2D, the kinetic energy dissipation rate mainly concentrates on the high gradient of velocity and small scale flow gradually polymerizes with time evolution at the section of half vertical height, the intermittency of temperature field in 3D cases are lower than that in 2D cases when the RT convection is developing, and the fluctuations in 3D cases are easier to dissipate than that of 2D cases.

This paper is organized as follows. In Section 2, the dynamics equations of the Rayleigh–Taylor flow and double lattice Boltzmann equations are briefly described. After that, the detailed results of kinetic and thermal energy dissipation rates with temporal evolution are discussed. Finally, some summarizing conclusions are represented. 

## 2. Dynamics Equation and Numerical Methods

In the following section, the dynamics equation of Rayleigh–Taylor flow and double lattice Boltzmann equations are introduced, respectively.

### 2.1. Convection Diffusion Equation of Thermal Fluid

The convection diffusion equation of thermal fluid is the classical Oberbeck–Boussinesq Equations (2) and (3), Their expression are as follows [6,8].
(3)$$\frac{\partial \rho }{\partial t}+\nabla \cdot (\rho u)=0,$$
(4)$$\frac{\partial (\rho u)}{\partial t}+u\cdot \nabla (\rho u)=-\nabla p+\nabla \cdot (2\rho \nu S)-g\beta \Delta \theta ,$$
(5)$$\frac{\partial \theta }{\partial t}+u\cdot \nabla \theta =\kappa \nabla^{2}\theta ,$$

In which ν denotes the kinematic viscosity, κ denotes the diffusivity, ρ denotes the density of fluid, u denotes the macroscopic velocity, θ is the macroscopic temperature, Δθ represents the temperature difference between top and bottom boundaries, β is the thermal expansion coefficient, g is the force of gravity, S is the stress term ($S=\frac{1}{2}(\frac{\partial u_{i}}{\partial x_{j}}+\frac{\partial u_{i}}{\partial x_{j}}),(i,=1,2,3)$), and p denotes the fluid pressure, respectively.

The finite element methods [25,26] and the finite volume method are effective tools used to solve some partial differential equations on complex geometries, which are applied to describe the complex fluid physical phenomena [27,28,29,30]. A large number of numerical methods are widely used to solve classical Oberbeck–Boussinesq equations [31,32,33,34,35,36]. The finite element methods [36], finite difference method [34] and the finite volume method [35] are traditional macroscopic methods for Computational Fluid Dynamics (CFD) calculation. The lattice Boltzmann method (LBM) is a computational fluid dynamics method based on a mesoscope simulation scale [37,38,39,40,41]. Compared with other traditional CFD calculation methods, this method has mesoscopic model characteristics between the micro molecular dynamics model and macro continuous model. LBM also has the advantages of simple description of fluid interaction, is widely easier to set complex boundary, easier to parallel calculation, easy to implement program and so on [42]. LBM has been widely considered as an effective method to describe fluid motion and deal with engineering problems [43]. In the next subsection, double distribution LBM for will be introduced.

### 2.2. Numerical Method for Rayleigh–Taylor Flow Equation

In the past thirty years, the lattice Boltzmann method (LBM) has increasingly risen to be an effective tool to simulate complex fluid [41,42,43,44,45,46,47,48,49,50,51,52]. LBM has very low numerical dissipation and dispersion errors [42,44], and is especially suitable for numerical calculation of incompressible fluids [41,42]. To describe the mass and dynamics macroscopic phenomena of the classical Oberbeck–Boussinesq equation, the so-called lattice Boltzmann equation is implemented by second-order multiscale expansion. The lattice Bhatnagar–Gross–Krook (LBGK) model for the fluid flow field [42]:(6)$$f_{i}(x+c_{i}\Delta t,t+\Delta t)=f_{i}(x,t)+(f_{i}^{eq}(x,t)-f_{i}(x,t))/\tau_{\nu }+F_{i},$$

In which $f_{i}(x,t)$ is the density distribution function, at (x,t), $c_{i}$ is the discrete velocity, $F_{i}$ denotes the discrete force term in Equation (6), ($F_{i}=3w_{i}\cdot \rho \cdot g\cdot \beta \cdot \theta \cdot c_{iz}$) [42] and $\tau_{\nu }$ denotes the relaxation times for density evolution equation in the lattice Boltzmann equation. The equilibrium function for the density distribution is represented by the following equation [41].
(7)$$f_{i}^{eq}=\rho w_{i}[1+\frac{c_{i}\cdot u}{c_{s}^{2}}+\frac{{\left(c_{i}\cdot u\right)}^{2}}{c_{s}^{2}}-\frac{u^{2}}{2c_{s}^{2}}],$$
where $w_{i}$ represents the weight coefficient in the D3Q19 [41], the D3Q19 lattice is implemented in this paper, i is 0,1,⋯18 and ν denotes the kinematic viscosity in mesoscopic method by the following expression.
(8)$$\nu =\frac{2\tau_{\nu }-1}{6}\frac{{\left(\Delta x\right)}^{2}}{\Delta t},$$

More detailed informations regarding the LBM discretization are introduced. Two separate parts (streaming and collision) can be identified by having a close look at Equation (6). One comes from the integration along characteristics $f_{i}(x+c_{i}\Delta t,t+\Delta t)-f_{i}(x,t)$. The other comes from the operator of local collision $(f_{i}^{eq}(x,t)-f_{i}(x,t))/\tau_{\nu }+F_{i}$. The LBGK equation into distinct streaming (or propagation) and collision steps can be logically separated. Overall, at time *t* and point *x*, each lattice site stores *q* populations $f_{i}$. Each population $f_{i}(x,t)$ receives a collisional contribution in the collision step or relaxation step and each population $f_{i}(x,t)$ can become
(9)$$f_{i}^{*}(x,t)=f_{i}(x,t)+(f_{i}^{eq}(x,t)-f_{i}(x,t))/\tau_{\nu }+F_{i},$$

The collision can be a purely algebraic and local operation. The $f_{i}^{*}(x,t)$ is the population state after collision. The other step is the streaming or propagation step. Here, the post-collision populations $f_{i}^{*}(x,t)$ just stream along their associated direction ci to reach a neighbouring lattice site where they become $f_{i}(x+c_{i}\Delta t,t+\Delta t)$. A non-local operation is given as $f_{i}(x+c_{i}\Delta t,t+\Delta t)=f_{i}^{*}(x,t)$. The memory content of $f_{i}^{*}(x,t)$ is copied to the lattice site at $x+c_{i}\Delta t$ and it old lattice information is overwritten. One common strategy is to use two sets of populations, one for reading data, the other for writing data.

To solve convective diffusion equation of the classical Oberbeck–Boussinesq equation, the lattice Boltzmann equation for the temperature field is given by the following equation [43].
(10)$$g_{i}(x+c_{i}\Delta t,t+\Delta t)=g_{i}(x,t)+(g_{i}^{eq}(x,t)-g_{i}(x,t))/\tau_{\theta },$$
where $g_{i}(x,t)$ is temperature distribution function, $\tau_{\theta }$ denotes the relaxation times for temperature evolution equation.The equilibrium function of the temperature distribution is represented by the following expression [48].
(11)$$g_{i}^{eq}=\theta w_{i}[1+\frac{c_{i}\cdot u}{c_{s}^{2}}+\frac{{\left(c_{i}\cdot u\right)}^{2}}{c_{s}^{2}}-\frac{u^{2}}{2c_{s}^{2}}]$$
where the diffusivity number κ is represented in mesoscopic method by the following equation.
(12)$$\kappa =\frac{2\tau_{\theta }-1}{6}\frac{{\left(\Delta x\right)}^{2}}{\Delta t},$$

More detailed informations regarding the discretization of Equation (10) is similar to the Equation (6). The Macroscopic density, velocity, and temperature are obtained by calculating the mesoscopic variables.
(13)$$\rho =\sum_{i=0}^{8}f_{i},\;\rho u=\sum_{i=0}^{8}c_{i}f_{i},\;\theta =\sum_{i=0}^{8}g_{i},$$

The expressions of density, momentum, and temperature are derived by leading into a Chapman–Enskog expansion [42]. Lattice Boltzmann equation (Eqations (6) and (10)) can be derived by expansion of a spatial scale (x_1_ = ε*_x_*) and two time scales (*t*_1_ = ε*_t_*, *t*_2_ = *ε_t_*) to respectively obtain the classical Oberbeck–Boussinesq equations (Equations (1), (2) and (3)) using the above Chapman–Enskog expansion [42].

The *Rayleigh* number (*Ra*) is a critical non-dimensional number in the RT convection. The *Ra* is defined as
(14)$$Ra=\frac{\beta \Delta \theta gL_{z}{}^{3}}{\nu \kappa },$$

The enhancement of the heat transfer can be calculated by the *Nusselt* number in LBM [43,48].
(15)$$Nu=1+\frac{\langle u_{z}\theta \rangle }{\kappa \Delta \theta /L_{z}},$$
where $u_{z}$ represents the vertical velocity, Δθ denotes the temperature difference between the up and bottom boundaries, $L_{z}$ is the computational height, and 〈.〉 is the average over the whole flow domain.

A computational domain size (1×1×2) is defined by $L_{x}\times L_{y}\times L_{z}$ with the resolutions 256×256×512 for $Ra=1.8\times 10^{9}$. The no-slip condition is implemented in the all boundaries for all numerical simulations. The turbulent RT mixing instability is seeded by giving the initial condition of the unstable step profile. The perturbations of sinusoidal wave are executed in order to drive the independence of the turbulent state from initial conditions [15]. At the beginning of the system, single fluid of two temperatures in all physical system is at rest and the hotter uniform fluid layer (θ=1) is placed on bottom of the colder one (θ=−1) with an initial jump of temperature $\Theta_{0}$ ($\Theta_{0}=\theta_{2}-\theta_{1}$). These corresponding parameters are introduced in the computational initial conditions of all system.The time dependent turbulence with initial condition is u(x,0)=0, $\theta (x,0)=-(1/2)\theta_{0}\mathrm{sgn}(z)$, and the Atwood number $A=\beta \Theta_{0}/2$, a superposition of cosine waves of wave numbers 16≤ *k* ≤ 32 is performed in the initial temperature interface *θ* = 0 at *z* = 1. For repeat-ability purposes, a total of 16 independent realization evolutions of the 3D RT mixing zone have been performed by adding different perturbed interfaces. In all the numerical simulations, *Ag* = 0.25, *L_z_* = 2, and *Θ*_0_ = 2, (the corresponding *Prandtl* number is *Pr* = *ν/κ* = 1). The fluctuation of mixing velocity linearly grows with time evolution [1,11] and the mixing layer width grows non-linearly in time [1,13].

## 3. Results and Discussion

In this section, first of all, the temperature fields, the logarithmic kinetic energy dissipation rates and the logarithmic thermal energy dissipation rates are presented with time evolution and mixing central section. In addition, the probability density functions of kinetic energy dissipation rates and thermal energy dissipation rates are obtained in the 3D turbulent RT mixing zone. Moreover, the fractal dimension of mixing interface is displayed. Finally, the spatial scaling laws of velocity and temperature structure functions are given, respectively.

### 3.1. Profiles of $\varepsilon_{u}$ and $\varepsilon_{\theta }$ in 3D Case

In order to reveal the statistical properties of the temporal–spatial evolution of kinetic and thermal energy dissipation rates in turbulent Rayleigh–Taylor mixing, the temperature fields, logarithmic kinetic and thermal energy dissipation rate with time evolution will be represented in the 3D case and the 2D section of *L_z_*/2. During the time evolution of the Rayleigh–Taylor mixing zone, the temperature mixing layer grows into a complex geometrical object characterized by plumes and entertainment regions.

Figure 1 illustrates the cold-hot fluid mixing process of the the temperature fields with time evolution at times t/τ=2,3 and 4.5. Here, ***τ*** denotes the characteristic time ($\tau =\sqrt{L_{z}/Ag}$) [11,20]. Plotted in Figure 1, we can clearly see that with time evolution, the mixed region of hot or cold fluid begins to increasingly expand the opposite region, a large number of small scale thermal plumes initially emerge, are passively transported and gradually polymerized thermal plumes of large scale in the interface of cold-hot temperature, and large-scale nonlinear phenomena is characterized by the formation of descending and ascending plumes, which enhances the heat transport between the two reservoirs. The above discussions are also theoretically predicted [7], and are indeed observed in numerical previous studies [12,16].

Figure 2 describes the logarithmic kinetic energy dissipation rate in the 3D case with time evolution at times t/τ=2,3 and 4.5. The kinetic energy dissipation rate mainly occurs at the high zone of mixing strength, and expands with increasing mixing zones. Figure 3 displays the logarithmic thermal energy dissipation rate of global quantities with temporal evolution. As displayed in Figure 3, it is clearly observed that the thermal energy dissipation rate concentrates on the high gradient of temperature, which is well consistent with the theoretical assumption according to Equation (2). From the compare between Figure 2 and Figure 3, we can obtain that the thermal energy dissipation rate plays a dominant role in the mixing process of cold–hot temperatures, which further demonstrates the previous study in the 2D case [12].

In the 2D case, the kinetic and thermal dissipation rates with only one plane occur in the turbulent RT mixing evolution. In order to reveal the more spatial properties with time evolution in 3D, the global qualities of section (*z* = *L_z_*/2) will be provided. Figure 4a displays typical snapshots of the instantaneous temperature field obtained at three different times in the RT mixing evolution, at section (*z* = *L_z_*/2) and times t/τ=2,3 and 4.5. The corresponding velocity, log-scale kinetic and thermal energy dissipation rates are also displayed in Figure 4b–d. As displayed in Figure 4a, one can clearly see that small scale thermal plumes initially at section of H/2 increasingly develop large scale plumes with time evolution. The comparison between Figure 4b and Figure 4c indicates that the kinetic energy dissipation rate mainly concentrates on the high gradient of velocity and small scale flow gradually polymerizes with time evolution, which agrees with the theoretical assumption according to Equation (1). The comparison between Figure 4a and Figure 4d suggests that the thermal energy dissipation rate mainly concentrates on the high gradient of temperature with temporal evolution, which is also consistent with the theoretical assumption according to Equation (2).

In order to reveal the statistical properties of both the kinetic energy dissipation rate and thermal energy dissipation rate in mixing zone, we mainly focus on the vertical profiles of averaged kinetic ${\langle \varepsilon_{u}\rangle }_{\left(x,y\right)}$ and thermal energy dissipation rates ${\langle \varepsilon_{\theta }\rangle }_{\left(x,y\right)}$ on the horizontal plane. Chertkov [6] proposed a theoretical model based on the “5/3”-K41 scenario for velocity and temperature spectra in 3D turbulent Rayleigh–Taylor mixing zone. Its theoretical model in 3D is followed as [13].
(16)$${\langle \varepsilon_{u}\rangle }_{V}∼t\;\mathrm{and}\;{\langle \varepsilon_{\theta }\rangle }_{V}∼t^{-1},$$

Nevertheless, its theoretical model in 3D is very different from that in 2D, which is mainly based on the BO59 scaling. Its theoretical expression in 2D is followed as [6,11,12]
(17)$${\langle \varepsilon_{u}\rangle }_{V}∼t^{-0.5}\;\mathrm{and}\;{\langle \varepsilon_{\theta }\rangle }_{V}∼t^{-1},$$
where ${\langle \ldots \rangle }_{V}$ represents a spatial average inside the turbulent RT mixing zone. The ${\langle \varepsilon_{u}\rangle }_{V}$ and ${\langle \varepsilon_{\theta }\rangle }_{V}$ decrease with time evolution in velocity and temperature spectra in 3D RT system. Figure 5 describes the vertical profiles of averaged kinetic energy dissipation rate at times t/τ=2,3,4 and 4.5.

As displayed in Figure 5, we can see that the vertical profiles of averaged kinetic energy dissipation rate increases with time evolution of turbulent RT mixing zone, which is in qualitative agreement with the theoretical scaling relationships of the kinetic energy dissipation rate in the 3D case (See Equation (16)). Zhou et al. [12] had demonstrated that the amplitude of averaged kinetic energy dissipation rate in 2D situations is expected to decrease with time evolution according to the theoretical scaling (Chertkov (2003)) [6]. The total kinetic-energy dissipation increases with time evolution as a function of time. It is known that RT turbulence mainly shows an instance of the general case of a turbulent flow adiabatically a time-dependent evolution of energy.

Figure 6 illustrates the vertical profiles of averaged thermal energy dissipation rate at times t/τ=2,3,4 and 4.5. As shown in Figure 6, it is clearly observed that the amplitude of averaged thermal energy dissipation rate decreases with time evolution of turbulent RT mixing zone, which qualitatively agrees with the theoretical scaling relationships of the thermal energy dissipation rate in the 3D case (See Equation (16)). Zhou et al. [12] studied that the amplitude of averaged thermal energy dissipation rate in 2D situations decreases with time evolution according to the theoretical scaling (Chertkov (2003)) [6]. The total kinetic energy dissipation decreases with time evolution as a function of time. The vertical profiles statistical properties of both the kinetic energy dissipation rates and thermal energy dissipation rates can provide insight into understanding statistical properties in the turbulent RT mixing zone.

The comparison between the vertical profiles of averaged kinetic energy dissipation rate in Figure 5 and the vertical profiles of averaged thermal energy dissipation rate in Figure 6 shows that the ratio between thermal energy dissipation rate versus kinetic energy dissipation rate is about four orders of magnitude, which indicates that the thermal energy dissipation rate still dominates in the transport of thermal energy, the values of kinetic energy dissipation rate can almost be neglected compared to the growth rate of the total kinetic energy of the system [12]. However, the ratio between thermal energy dissipation rate versus kinetic energy dissipation rate evidently increases two orders of magnitude compared to the previous studies of kinetic and thermal energy dissipation rates in 2D [12]. This can be due to six directions velocity gradient of space and temperature gradient of space inside the RT mixing zone in the 3D case rather than two directions velocity gradient of space and temperature gradient of space inside the RT mixing zone in the 2D case.

### 3.2. Probability Density Functions of $\varepsilon_{u}$ and $\varepsilon_{\theta }$ in the 3D Turbulent RT Mixing Zone

The probability density function (*PDF*) is an effective approach to study the influence of small-scale intermittency on acceleration statistics [10]. Interestingly, the scalar dissipation rate is mainly focused on determining turbulent mixing zone. This is due to the fact that the scalar field and the dissipation field acquire larger amplitudes with increasing *Ra*. The lower panels show the scalar PDF of integrating by Chertkov et al. (1999) [7], where the modelling of thin reactive layers called flame-lets, are advected and embedded in the turbulent RT mixing zone. The stretched exponential function is followed as [12]
(18)$$P(Z)=\frac{C}{\sqrt{Z}}\exp(-nZ^{\alpha }),$$
where C, n, and α denote the parameters of fitting, and $Z=X-X_{mp}$, with $X=\varepsilon_{u}/{\left(\varepsilon_{u}\right)}_{rms}$ or $X=\varepsilon_{\theta }/{\left(\varepsilon_{\theta }\right)}_{rms}$, and $X_{mp}$ represents the most probable amplitude abscissa, respectively. In 3D case, the above fitting parameters in the fitting process are given as *n* = 0.90 and *α* = 0.82 for *ε_u_* and *n* = 1.13 and *α* = 0.66 for *ε_θ_* according to be identified and discussions of the previous studies [13]. To further reveal the energy dissipation features of the mixing zone, we will extend to the *PDF*s of the kinetic and thermal energy dissipation rates in 3D case. Figure 7 illustrates different *PDF* of the kinetic energy dissipation rate at times t/τ=2,3 and 4.5. Since the *Ra* is the same for the three different time evolution, the log scale PDF of the kinetic energy dissipation rate dividing their corresponding mean values is plotted to reveal the differences in 3D case. As illustrated in Figure 7, one can see that the self-similarity of velocity fluctuation is demonstrated by observing the *PDF* of *ε_u_* at distinct times collapse well on top of each *ε_u_*. Furthermore, the long tails of *PDF* of *ε_u_* reveal strong fluctuation, which is well consistent with the previous studies of both passive and active scalars [12,21].

Figure 8 displays the log scale *PDF* of the thermal energy dissipation rate with time evolution at t/τ=2,3 and 4.5 in the mixing zone. Plotted in Figure 8, it is obviously observed that the self-similarity of temperature fluctuation is indicated by observing the *PDF* of *ε_θ_* at distinct times collapse well on top of each *ε_θ_*. Meanwhile, the long PDF tail of *ε_θ_* reveal strong temperature fluctuation. Different from previous study [12], it is found that the PDF tail of *ε_θ_* is lower than the stretched exponent function. With the evolution of time, the simulated PDF tail of *ε_θ_* gradually rises and approaches the stretched exponent function. It indicates that the intermittency of temperature field in 3D cases are lower than that in 2D cases when the RT convection is developing. When it is fully developed, i.e., *t*/*τ*≥4, the intermittency of temperature field also fits the stretched exponent function.

### 3.3. Fractal Dimension of Mixing Interface

More detailed dissipation events occur in the colder–hotter layer mixing interface of a single phase fluid. To further reveal the generation mechanisms of the kinetic and thermal energy dissipation rates with time evolution in the mixing zone, the fractal features of these interfaces are mainly focused in 3D space. Zhou et al. [12] studied the fractal properties of these interfaces with temporal evolution in 2D space. They argued that no constant fractal dimension occurs in the interfaces. In general, the box-counting method is widely used to study the fractal dimension of these mixing interfaces [12]. This is due to that during the RT evolution, the mixing zone expands.

The fluid interfaces between hot and cold fluid become increasingly complex. To obtain whether there is a constant fractal dimension for the interfaces between hot and cold fluid. In this paper, due to six directions velocity gradient of space and temperature gradient of space inside the RT mixing zone in the 3D case, we also continue to adopt the previous box-counting method to provide some fundamental understanding of the fractal properties of these fluid interfaces.
(19)$$d_{B}=\frac{\partial \mathrm{lg}N_{2}(r)}{\partial \mathrm{lg}(r)},$$
where *d_B_* is nearly unity, and *N*_2_ denotes the counted number. Figure 9 describes the number of square boxes *N_r_* of size *r* that mainly overlap the interfaces (the contours of *θ* = 0) versus the normalized box size *r*/*H* (*H*, the longitudinal height of computational area) on a log-log scale at different times *t*/*τ* = 1.5, 2, 3, 4, and 4.5. As shown in Figure 9, it is clearly observed that *N_r_* increases with time evolution for all the scale studies, the fractal dimension is a little higher than 2 in a range scale at time *t*/*τ* = 1.5, and the fractal dimension for the fluid interfaces is 2.4 in a range scale at times *t*/*τ* = 2, 3, 4 and 4.5, which indicates that there is a constant fractal dimension for the interfaces between hot and cold fluid during a certain scale and a value of 2.4 is obtained for the iso-surfaces of thermal plumes near the interfaces between hot and cold fluid. This is consistent with that for active scalar obtained in another buoyancy-driven turbulence, where a constant fractal dimension with some values of 1.50 ± 0.02 were obtained for the iso-surfaces of thermal plumes [11]. As reported in previous 2D study by Zhou [12], the fractal dimensions are similar at *t*/*τ* ≥ 2, which indicates the self-similarity of the turbulent RT mixing in the turbulent range. Different from 2D cases, it observed that the fractal dimension is constant (2.4) at *t*/*τ* ≥ 2 in our simulations. This difference between 2D and 3D cases is caused by different dimensions of the simulated fluctuation inside the RT mixing zone. The fluctuations in 2D cases are more difficult to dissipate than in 3D cases. Therefore, at large scale, the fractal dimension in 2D simulations increases. This further demonstrated that the mixing zone in 3D space expands, the fluid interfaces between hot and cold fluid become increasingly complex in 3D space, and the self-similarity of the fluid mixing zone becomes increasingly prominent during the turbulent range.

### 3.4. Spatial Intermittency in Mixing Zone

The theory of a mixing layer in the self-similar regime is proposed by Chertkov (2003) [6], which is based on the Kolmogorov–Obukhov adiabatic generalization of steady Navier–Stokes turbulence (Kolmogorov 1941,Obukhov 1941) [22]. In order to reveal the statistical properties of spatial intermittency, the scaling relationships of the velocity and temperature structure functions in the 3D case are followed as [21]
(20)$$S_{p}^{V}(r)=\langle {\left[\left(u(r,t)-u(0,t)\right)\cdot \frac{r}{r}\right]}^{p}\rangle ≃{\left(g\beta \theta_{0}\right)}^{2p/3}t^{p/3}r^{p/3},$$
(21)$$S_{p}^{\theta }(r)=\langle {\left[\theta (r,t)-\theta (0,t)\right]}^{p}\rangle ≃\theta_{0}^{p}{\left(g\beta \theta_{0}\right)}^{-p/3}t^{-2p/3}r^{p/3},$$
where, p denotes the order of structure function. The theory of turbulent fluctuation in 2D case that is the B59 scenario essence have been proposed by Chertkov (2003) [6]. Boffetta et al. (2009) [9], and Biferale et al. (2010) [19] investigated the validity of the Bolgiano–Obukhov scenario by DNS in 2D and using a thermal LBM, respectively. Zhou (2013,2016) [11,13] and Qiu et al. (2014) [14] studied B59 scaling of the velocity and temperature structure functions both in 2D space and in time by DNS and mainly considered longitudinal velocity and temperature structure functions over horizontal separations. Figure 8 and Figure 9 illustrate a log-log plot of the velocity structure function and temperature structure function of orders p=2, p=4 and p=6 in the mixing zone, respectively. As shown in Figure 8, it is clearly observed that the second-order velocity structure function $S_{2}^{V}(r)$ displays a linear scaling in a range of 7≤r/η≤20, the fourth-order velocity structure function $S_{4}^{V}(r)$ shows a linear scaling in a range of 9≤r/η≤29, the sixth-order velocity structure function $S_{4}^{V}(r)$ represents a linear scaling in a range of 7≤r/η≤22, which is well consistent with the previous theoretical prediction [6], where r denotes the space scale and η represents the Kolmogorov scale [11]. We further note that this feature is qualitatively consistent with those observed in 3D cases [16,19]. Here, η is the Kolmogorov scale. Structure functions are computed by taking differences of the second components of the velocity in the *r* = *x*,and *y* direction to reveal the powerful anisotropy and inhomogeneity of turbulent RT mixing flow, again averaging over the central and over the y-direction part of the mixing layer. 

Plotted in Figure 9, one can see that the second-order temperature structure function $S_{2}^{\theta }(r)$ indicates a linear scaling in a range of 7≤r/η≤20, the fourth-order temperature structure function $S_{4}^{\theta }(r)$ represents a linear scaling in a range of 9≤r/η≤29, the sixth-order temperature structure function $S_{4}^{\theta }(r)$ displays a linear scaling in a certain range of 7≤r/η≤22, which is well consistent with the previous theoretical prediction [6]. Although a *log*-*log* scaling the global overall is in agreement between present results and dimensional Bolgiano scaling, important deviations can be obtained both at the crossover between viscous and inertial range. Figure 10 and Figure 11 also demonstrate that in the mixing zone, the kernels of integration are extraordinarily asymmetric for both velocity and temperature, a linear signature of persistence of cliff-ramp-like structures of the velocity and temperature fields, like fronts of plumes/spikes [9,11,21].

## 4. Conclusions

In this paper, an investigation into the temporal–spatial evolution of kinetic and thermal energy dissipation rates in three-dimensional turbulent Rayleigh–Taylor mixing has been performed using the lattice Boltzmann method. Some key conclusions are as follows:

First of all, small scale thermal plumes initially at section *H*/2 increasingly develop large-scale plumes with time evolution. The kinetic energy dissipation rate is mainly concentrated in areas of large velocity gradients and small-scale flow gradually polymerizes with time evolution, and the thermal energy dissipation rate is mainly concentrated in zones with large temperature gradients, which is consistent with the theoretical assumption.

Moreover, the thermal energy dissipation rate mainly dominates in the transport of thermal energy, and the values of the kinetic energy dissipation rate can almost be neglected compared to the growth rate of the total kinetic energy of the system in the mixing zone. The ratio between thermal energy dissipation rate versus kinetic energy dissipation rate evidently increases two orders of magnitude in the 3D case compared to the previous studies of kinetic and thermal energy dissipation rates in 2D cases.

In addition, the PDF tail of *ε_θ_* is lower than the stretched exponent function in the mixing zone which is different from previous studies in 2D cases. The simulated PDF tail of *ε_θ_* gradually rises and approaches the stretched exponent function with temporal evolution, which indicates that the intermittency of the temperature field in 3D cases is lower than that in 2D cases when the RT convection is developing to turbulence.

Furthermore, the slope of fractal dimension increases at early time evolution, however, the fractal dimension for the fluid interfaces is 2.4 at times *t*/*τ* ≥ 2 in the 3D turbulent RT mixing zone. The fluctuations in 3D cases are easier to dissipate than those in 2D cases. This is mainly due to the rising iso-surfaces of thermal plumes, which again demonstrates the self-similarity of the turbulent RT mixing in the turbulent range. Moreover, the real 3D turbulent RT mixing in the turbulent range are noticeably different from the 2D simulation prediction. The difference in slope between 2D and 3D can be caused by six directions velocity spatial gradient and temperature spatial gradient inside the RT mixing zone in the 3D case. This is further demonstrated that the mixing zone in 3D space expands, the fluid interfaces between hot and cold fluid become increasingly complex in 3D space, and the self-similarity of the fluid mixing zone becomes increasingly prominent during the turbulent range.

Finally, our numerical results reveal that the second, fourth and sixth-order structure functions for velocity and temperature possess a linear scaling over a certain range within the inertial range, which is consistent with the previous theoretical prediction.

The present study can only be a first step. A formal conclusion on the transitional behaviour of the kinetic and thermal energy dissipation rates requires data to consider the effect of rotation at larger Rayleigh numbers. The dependence on the dimensionless Rossby number will be studied in the follow-on work. This would also help us to better understand the parameter dependence of the kinetic and thermal energy dissipation rates in comparison to the other scales. In the present study, the kinetic and thermal energy dissipation rates are similar in magnitude, since the dimensionless Rossby number is close to zero, but for larger dimensionless Rossby numbers, the kinetic and thermal energy dissipation rates will differ significantly. In particular, for very high Rossby numbers, we can expect dramatic changes in the mixing layer dynamics and the related self-similar regime based on our current investigation in this direction, which will be discussed in our future work.

## Figures and Tables

**Figure 1 entropy-22-00652-f001:**
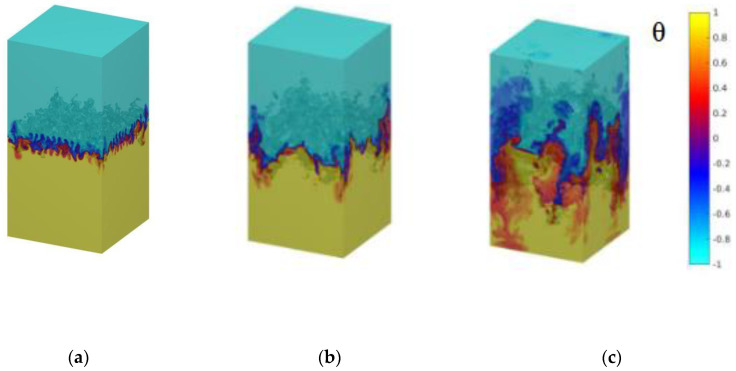
Temperature fields with time evolution at times and (**a**) *t*/*τ* = 2 (**b**) *t*/*τ* = 3, and (**c**) *t*/*τ* = 4.5.

**Figure 2 entropy-22-00652-f002:**
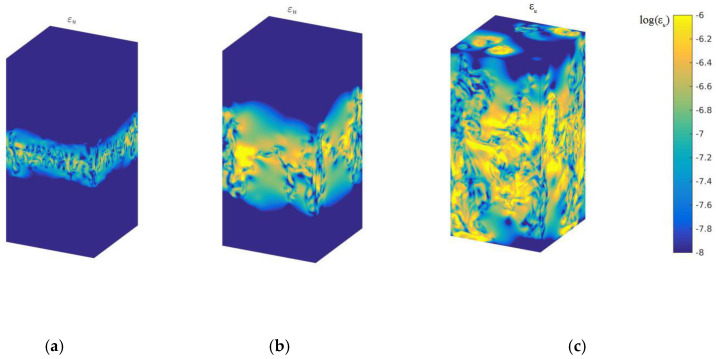
Logarithmic kinetic energy dissipation rate with time evolution, (**a**) *t*/*τ* = 2 (**b**) *t*/*τ* = 3, and (**c**) *t*/*τ* = 4.5.

**Figure 3 entropy-22-00652-f003:**
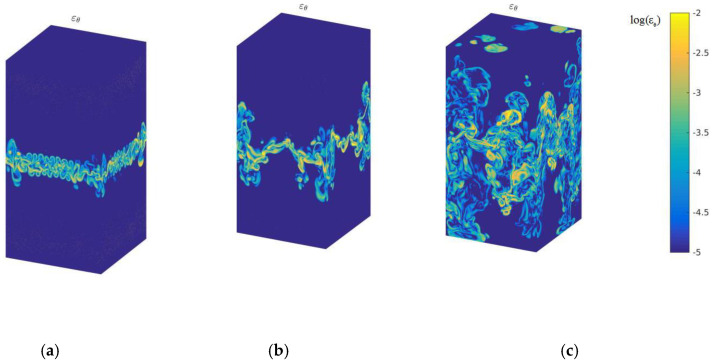
Logarithmic thermal energy dissipation rate with time evolution, (**a**) *t*/*τ* = 2 (**b**) *t*/*τ* = 3, and (**c**) *t*/*τ* = 4.5.

**Figure 4 entropy-22-00652-f004:**
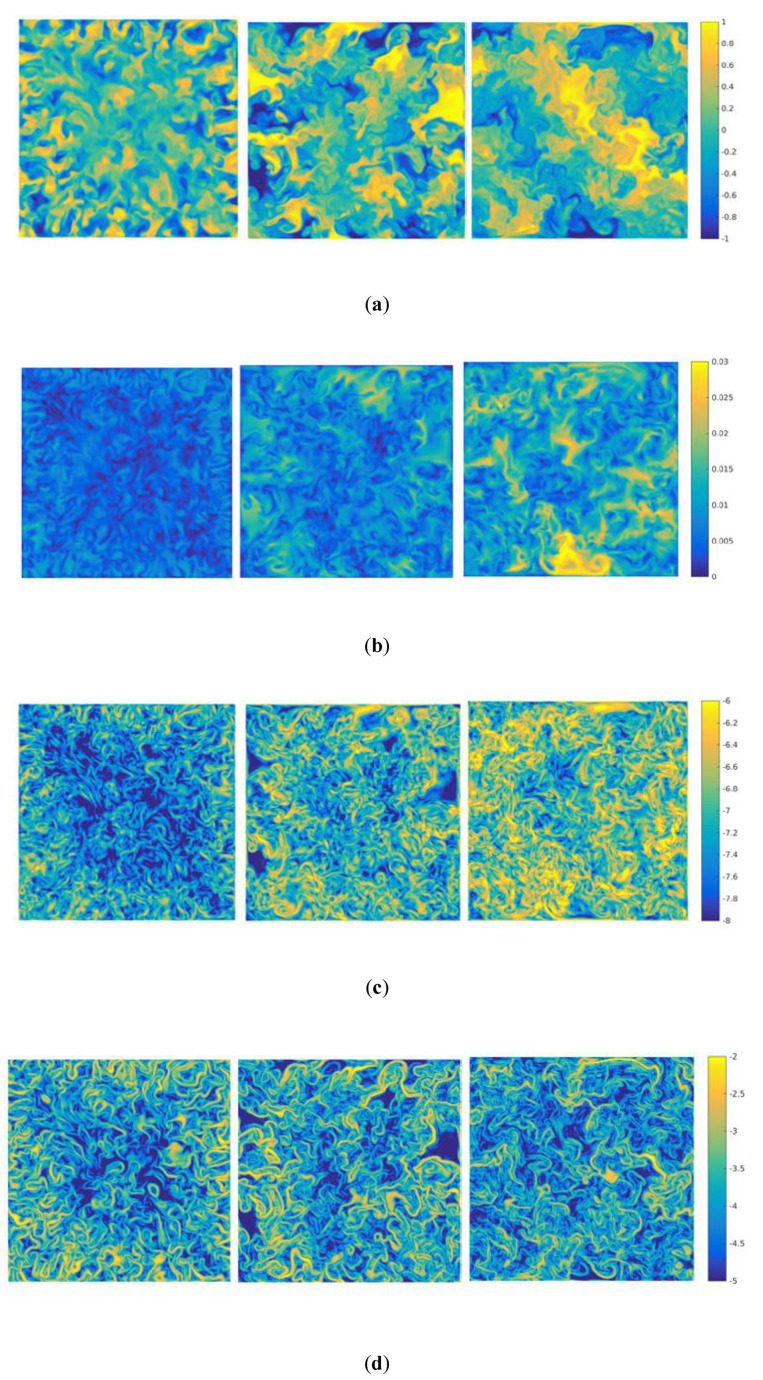
Time evolution of the temperature, velocity, logarithmic kinetic and thermal energy dissipation rates at section (z = *L_z_*/2) and times t/τ=2,3 and 4.5, (**a**) temperature, (**b**) velocity,(**c**) logarithmic kinetic energy dissipation rate, and (**d**) logarithmic thermal energy dissipation rate.

**Figure 5 entropy-22-00652-f005:**
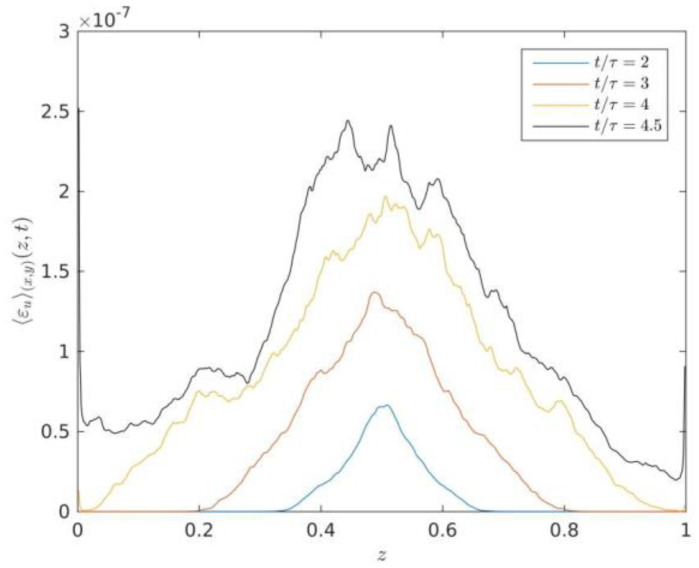
Vertical profiles of averaged kinetic energy dissipation rate at times t/τ=2,3,4 and 4.5.

**Figure 6 entropy-22-00652-f006:**
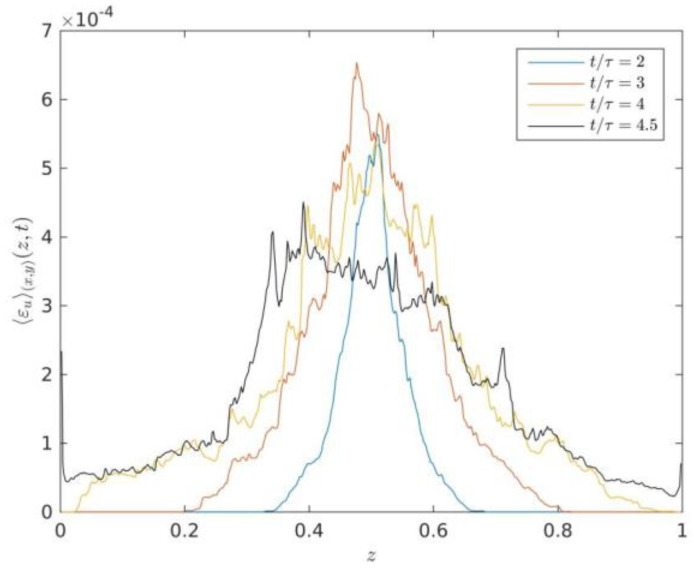
Vertical profiles of averaged thermal energy dissipation rate at times t/τ=2,3,4 and 4.5.

**Figure 7 entropy-22-00652-f007:**
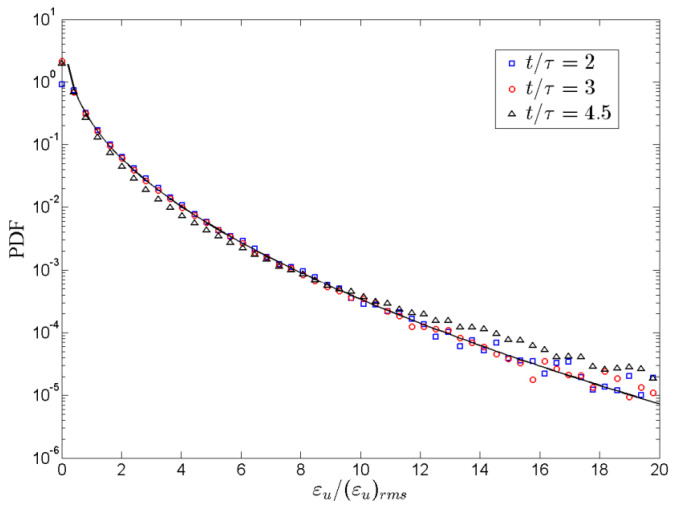
*PDF* of the kinetic energy dissipation rate in turbulent RT mixing zone at times t/τ=2,3 and 4.5 (root- mean-square (rms) where $i_{rms}=\sqrt{{\langle {\left(i-{\langle i\rangle }_{j}\right)}^{2}\rangle }^{2}}$ is the rms value of i).

**Figure 8 entropy-22-00652-f008:**
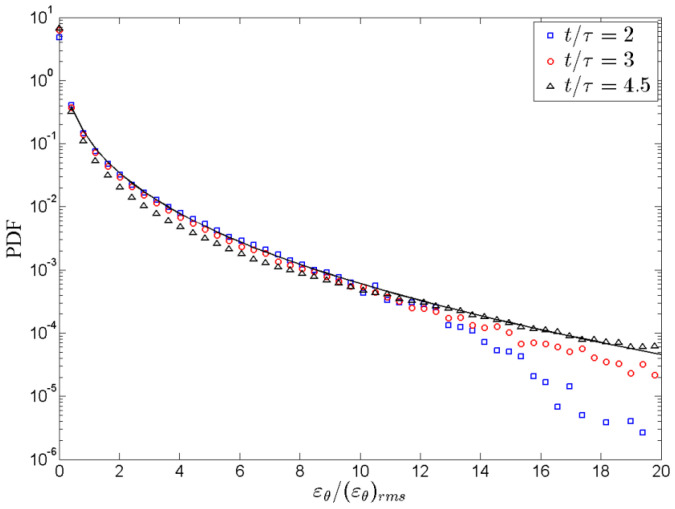
*PDF* of the thermal energy dissipation rate in turbulent RT mixing zone at times t/τ=2,3 and 4.5.

**Figure 9 entropy-22-00652-f009:**
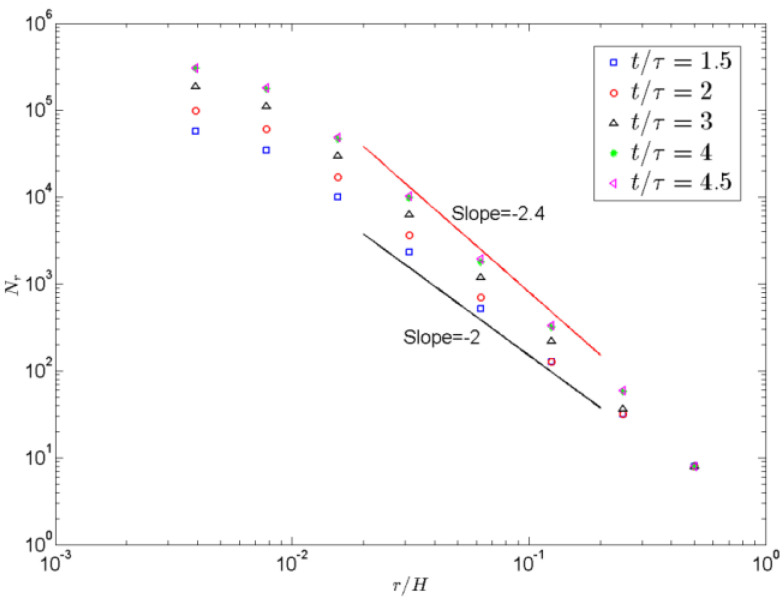
Number of square boxes *N_r_* of size r that mainly overlap the interfaces (the contours of θ = 0) versus the normalized box size r/H on a log-log scale at different times *t*/*τ*= 1.5, 2, 3, 4, and 4.5.

**Figure 10 entropy-22-00652-f010:**
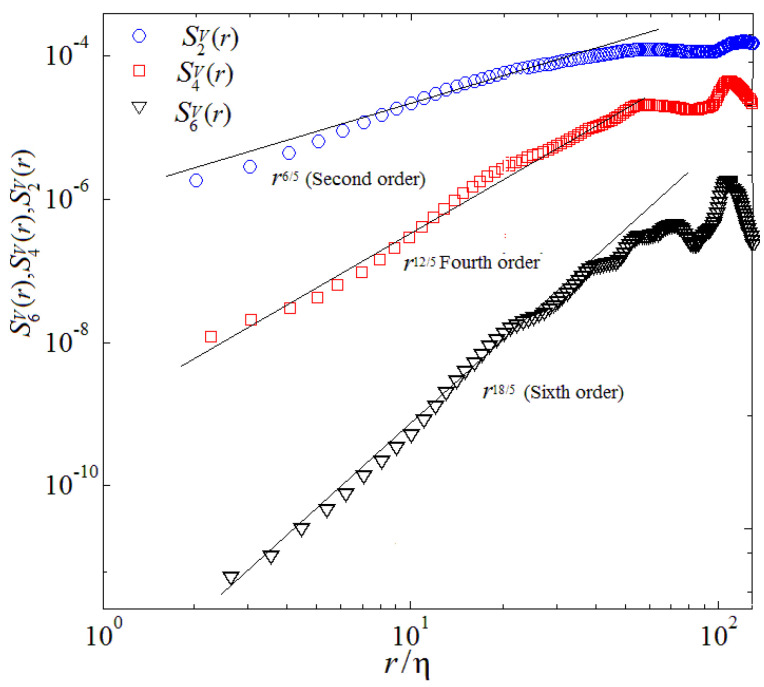
Velocity structure function of orders at *p* = 2, 4 and 6, (r/η denotes the dimensionless space scale).

**Figure 11 entropy-22-00652-f011:**
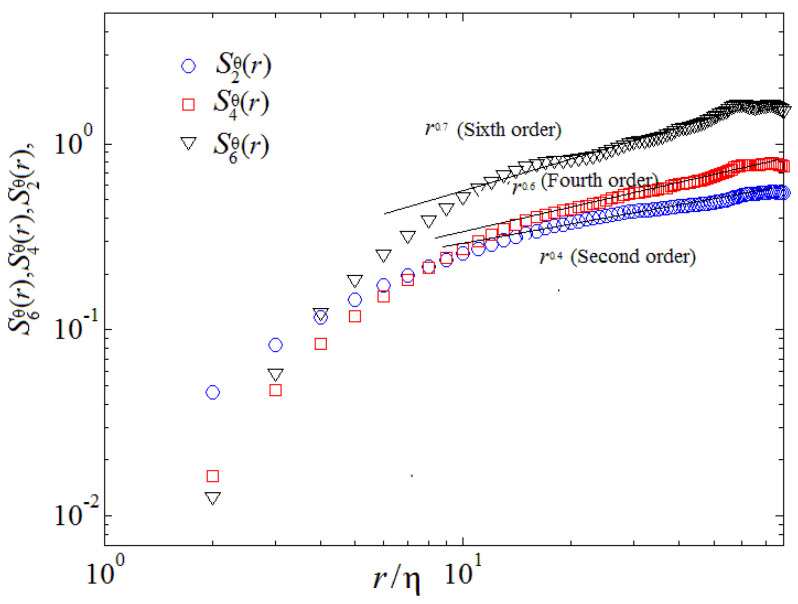
Temperature structure function of orders at *p* = 2, 4 and 6 (r/η denotes the dimensionless space scale).

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
