# Peer review of "Temporal–Spatial Evolution of Kinetic and Thermal Energy Dissipation Rates in a Three-Dimensional Turbulent Rayleigh–Taylor Mixing Zone"

_entropy, 2020, doi:10.3390/e22060652_

Round 1

Reviewer 1 Report

There are misprints: lines 124-126, 144, 293-294, 311, 367 and so on, that must be correct. Some text editing is needed.

Author Response

Response of entropy-810394

Comments and Suggestions for Authors

There are misprints: lines 124-126, 144, 293-294, 311, 367 and so on, that must be correct. Some text editing is needed.

Response: Corrected.

In sum, the authors appreciate sincerely the referees’ valuable comments and suggestions on this work.

Reviewer 2 Report

The manuscript submitted to Entropy falls within the subject of the journal. The authors make a good introduction to the topic clearly defining their main contribution. It is well organzed and structured, figures are relevant and well described. Conclusions are well supported by the results.  Overall, the work is good, however, it would have been good to have included a discussion section of the results to further explore the topic.

Author Response

Response of entropy-810394

Comments and Suggestions for Authors

The manuscript submitted to Entropy falls within the subject of the journal. The authors make a good introduction to the topic clearly defining their main contribution. It is well organzed and structured, figures are relevant and well described. Conclusions are well supported by the results.  Overall, the work is good, however, it would have been good to have included a discussion section of the results to further explore the topic.

Response: Corrected.

This is further demonstrated that the mixing zone in 3D space expands , the fluid interfaces between hot and cold fluid become increasingly complex in 3D space, and the self-similarity of the fluid mixing zone becomes increasingly prominent during the turbulent range.

 In sum, the authors appreciate sincerely the referees’ valuable comments and suggestions on this work.

Reviewer 3 Report

The authors provide a LBM resolution to the coupled transport equations. Specifically, the authors present a research about the temporal-spatial evolution of kinetic and thermal energy dissipation rates in three-dimensional turbulent Rayleigh-Taylor mixing. However, although the present work shows result that are of interest to the readers of the Entropy journal it is not acceptable in its actual form. Some issues that this reviewer suggest to fix, are, 

  • More detailed information regarding the mathematical model is required. For instance the stress term S Is not defined.
  • The figures are not properly labeled. It is hard to grasp on the meaning of the directions. More organization is required.
  • More detailed information regarding the LBM discretization employed. It is not clear how many integration points per lattice the authors are employing. 
  • More discussion regarding the fractal dimension and its impact in the dissipation.

Author Response

Response of entropy-810394

Comments and Suggestions for Authors

The authors provide a LBM resolution to the coupled transport equations. Specifically, the authors present a research about the temporal-spatial evolution of kinetic and thermal energy dissipation rates in three-dimensional turbulent Rayleigh-Taylor mixing. However, although the present work shows result that are of interest to the readers of the Entropy journal it is not acceptable in its actual form. Some issues that this reviewer suggest to fix, are,

1.More detailed information regarding the mathematical model is required. For instance the stress term S Is not defined.

Responses : Corrected

2.The figures are not properly labeled. It is hard to grasp on the meaning of the directions. More organization is required.

Responses : Corrected

  1. More detailed information regarding the LBM discretization employed. It is not clear how many integration points per lattice the authors are employing.

Responses : Corrected

More detailed informations regarding the LBM discretization are introduced. Two separate parts( streaming and collision) can be identified by having a close look at equation (6) . One comes from the integration along characteristics . The other comes from the  operator of local collision.  The LBGK equation into distinct streaming (or propagation) and collision steps can be logically separated. Overall, at time t and point x each lattice site stores q populations. Each population receives a collisional contribution in the collision step or relaxation step and each population  can become

.

The collision can be a purely algebraic and local operation. The is the population state after collision. The other step is the streaming or propagation step. Here, the post-collision populations  just stream along their associated direction ci to reach a neighbouring lattice site where they become.A non-local operation is given as .The memory content of  is copied to the lattice site at  and it old lattice information is overwritten. One common strategy is to use two sets of populations, one for reading data, the other for writing data.

  1. More discussion regarding the fractal dimension and its impact in the dissipation.

Responses : Corrected

 This is due to that during the RT evolution, the mixing zone expands and the fluid interfaces between hot and cold fluid become increasingly complex. To obtain whether there is a constant fractal dimension for the interfaces between hot and cold fluid.The fractal  dimension for the fluid interfaces is 2.4 in a range scale at times t/τ = 2,3,4 and 4.5, which indicates that there is a constant fractal dimension for the interfaces between hot and cold fluid during a certain scale and a value of 2.4 is obtained for the iso-surfaces of thermal plumes near the interfaces between hot and cold fluid. This is consistent with that for active scalar obtained in another buoyancy-driven turbulence,where a constant fractal dimension with some values of 1.50 ± 0.02 were obtained for the iso-surfaces of thermal plumes. This is further demonstrated that the mixing zone in 3D space expands , the fluid interfaces between hot and cold fluid become increasingly complex in 3D space, and the self-similarity of the fluid mixing zone becomes increasingly prominent during the turbulent range.

In sum, the authors appreciate sincerely the referees’ valuable comments and suggestions on this work.

Round 2

Reviewer 3 Report

This reviewer agree with the actual version of the manuscript and thus I recommend it for its publication.